# Emerging Role of Glioma Stem Cells in Mechanisms of Therapy Resistance

**DOI:** 10.3390/cancers15133458

**Published:** 2023-07-01

**Authors:** Frank Eckerdt, Leonidas C. Platanias

**Affiliations:** 1Robert H. Lurie Comprehensive Cancer Center of Northwestern University, Chicago, IL 60611, USA; 2Division of Hematology-Oncology, Department of Medicine, Northwestern University, Chicago, IL 60611, USA; 3Medicine Service, Jesse Brown VA Medical Center, Chicago, IL 60612, USA

**Keywords:** glioblastoma, glioma stem cells, plasticity, heterogeneity, therapy resistance, immune checkpoint

## Abstract

**Simple Summary:**

Glioblastoma is among the most lethal cancers in humans. Glioblastomas exhibit a striking heterogeneity, which is maintained by a highly plastic, adaptable population of glioma stem cells (GSCs). Glioblastomas inevitably recur as therapy-resilient tumors, and accumulating evidence indicates key roles for GSCs in recurrence and resistance. GSCs contribute to therapy resistance not only by promoting tumor heterogeneity, but also by modulating components of the tumor environment. This includes crosstalk with cells of the immune system, facilitating immune evasion of malignant cells. This review revisits the decades-long journey from the identification and characterization of GSCs to elucidating their contributions to glioblastoma malignancy and mechanisms of therapy resistance. Finally, the most recent insights in GSC biology are discussed, including novel strategies that specifically target GSCs in the age of immune-based strategies.

**Abstract:**

Since their discovery at the beginning of this millennium, glioma stem cells (GSCs) have sparked extensive research and an energetic scientific debate about their contribution to glioblastoma (GBM) initiation, progression, relapse, and resistance. Different molecular subtypes of GBM coexist within the same tumor, and they display differential sensitivity to chemotherapy. GSCs contribute to tumor heterogeneity and recapitulate pathway alterations described for the three GBM subtypes found in patients. GSCs show a high degree of plasticity, allowing for interconversion between different molecular GBM subtypes, with distinct proliferative potential, and different degrees of self-renewal and differentiation. This high degree of plasticity permits adaptation to the environmental changes introduced by chemo- and radiation therapy. Evidence from mouse models indicates that GSCs repopulate brain tumors after therapeutic intervention, and due to GSC plasticity, they reconstitute heterogeneity in recurrent tumors. GSCs are also inherently resilient to standard-of-care therapy, and mechanisms of resistance include enhanced DNA damage repair, MGMT promoter demethylation, autophagy, impaired induction of apoptosis, metabolic adaptation, chemoresistance, and immune evasion. The remarkable oncogenic properties of GSCs have inspired considerable interest in better understanding GSC biology and functions, as they might represent attractive targets to advance the currently limited therapeutic options for GBM patients. This has raised expectations for the development of novel targeted therapeutic approaches, including targeting GSC plasticity, chimeric antigen receptor T (CAR T) cells, and oncolytic viruses. In this review, we focus on the role of GSCs as drivers of GBM and therapy resistance, and we discuss how insights into GSC biology and plasticity might advance GSC-directed curative approaches.

## 1. Introduction

Glioma is the most common primary cancer of the central nervous system (CNS), with glioblastoma (GBM) representing the most aggressive form (grade 4 glioma) of this disease [1] (reviewed in [2]). GBM is the most commonly occurring malignant tumor in the brain and CNS (48.6% of malignant tumors) [3]. While selected subtypes of gliomas can be successfully treated, GBM is inevitably lethal in part due to its general resistance to therapeutic interventions. The current standard-of-care protocol was established ~18 years ago and is comprised of maximal surgical resection followed by radiotherapy and systemic chemotherapy, resulting in a median overall survival of only 14.6 months [4] (which may extend up to 20 months with the addition of novel strategies, such as tumor-treating fields (TTFs) [5]) and a five-year survival of 6.8% [6].

Stem cells are drivers of embryonic development, and similarities between developing embryonic tissue and cancer tissue have long been appreciated. These similarities were first described in the mid-1800s by Rudolf Virchow and later conceptualized by Julius Cohnheim and Franco Durante, hypothesizing that remnants of embryonic tissue remain in adult organs where they—when activated—can give rise to cancer, formulating the “embryonic rest theory of cancer” (reviewed in [7]). Based on these theories, cancers arise from embryonal-like progenitor cells that are present, and become activated, in the wrong places. Thus, a modern version of the embryonal rest theory of cancer is that malignancies arise from tissue stem cells in adults.

Almost all adult tissues exhibit normal tissue-specific stem cells that exert key roles in homeostasis and repair in many different tissues and hematopoietic lineages. Early evidence for the existence of such cells came from clonogenic cells surviving radiation that were able to repopulate the spleen and to differentiate into functional marrow [8,9]. Key properties of stem cells are their long-term self-renewal capacity and their ability to give rise to multiple cell lineages to ensure maintenance of specific tissue function [10]. These developmental programs re-emerge in cancer stem cells (CSCs) to support the maintenance and progression of most cancers.

Initial evidence for the CSC hypothesis came from early transplantation experiments in leukemic cells, demonstrating that a single cell could initiate a cancer graft [11]. Renewed interest emerged when additional studies demonstrated that cancer can be initiated and maintained by a small fraction of cells that exhibit stem cell properties [12,13,14]. Since then, additional studies have corroborated the existence of stem-like cancer cells with tumor-initiating capabilities in solid tumors, such as breast [15], colon [16], esophagus [17], liver [18], lung [19], ovarian [20], prostate [21], stomach [22], thyroid [23], and brain [24,25], among others (reviewed in [26]).

In recent years, increasing evidence has suggested tumors depend on CSCs not only for tumor initiation and maintenance, but also for tumor heterogeneity [27,28,29,30,31,32,33,34,35,36,37,38,39]) and plasticity [40,41], both of which are critical contributors to therapeutic resilience. Evidence from GBM xenografts suggested that radiation triggers an increased percentage of glioma stem cells (GSCs) and more aggressive tumors upon serial transplantation [42]. It seems apparent that cells expressing pluripotency markers exist in gliomas, and these cells depict a higher tolerance for radiation as compared to other cells of the same tumor. When GSCs are maintained in a quiescent state, they are more resistant to treatment [43,44]. However, quiescent GSCs can give rise to, or coexist with, proliferative GSCs in a dynamic fashion [45], indicating that certain GSC subsets may exhibit more proliferative potential than previously assumed [35,41]. While many questions still remain, there is strong evidence for key roles of GSCs in therapy resistance and tumor recurrence. In this review, we focus on the characteristics of GSCs, their functional relevance for therapy resistance, and their potential as therapeutic targets for GBM.

## 2. Cancer Stem Cells in Glioblastoma

The first report describing the isolation of brain CSCs from primary human brain tumors (including GBM and medulloblastoma) employed cell sorting of CD133^+^ tumor cells that were able to generate neurospheres, displayed self-renewal capacity, multipotency, and high proliferation potential [24]. A subsequent study demonstrated as few as 100 CD133^+^ cells were sufficient to initiate tumors in immunodeficient mice, phenotypically similar to the patients’ original tumors, revealing a striking tumorigenicity of these CD133^+^ brain CSCs [25]. CD133 is a five-transmembrane cell-surface protein encoded by *PROM1* that is expressed in neural stem cells [46]. Based on these findings, the CD133 cell surface marker has attracted attention as a neural precursor marker, as well as a potential marker for tumor initiating cells. However, subsequent studies suggested CD133 is not an exclusive marker for brain CSCs because, in some GBM samples, CD133^−^negative cells are also able to initiate tumors [47,48,49]. Additionally, CD133^+^ and CD133^−^ GSCs have been shown to be able to convert into each other within one GBM [49,50]. Further GSC surface markers were identified, such as CD44 [51,52], CD15 [53], A2B5 [54,55] CD90 [56], integrin alpha-6 [57], and CD171/L1CAM [58]. Additional (intracellular) GSC markers have been appropriated from normal neural stem cells, such as BMI1 [59], MSI1/2 [59], Nanog [60], Nestin [61], and Sox2 [59], and, most recently, YAP/TAZ [62], among others [52,63]. Remarkably, a core set of four transcription factors (OCT3, SOX2, SALL2 and OLIG2) was sufficient to reprogram differentiated GBM cells into “induced” GSCs that recapitulate GSC properties and thus are likely to drive GSC maintenance [64]. Although the identification of these markers that are expressed in brain CSCs has greatly advanced our understanding of different cell populations with divergent self-renewal and tumor initiating capacity within a heterogeneous GBM, they provide only limited applicability to describe GSCs. Instead of identifying markers, the field is redirecting the focus toward these cells’ contributions to tumorigenic processes and their ability to self-renew and generate differentiated progeny. Thus, the term GSC should only be designated to cell populations that have been functionally validated for their capability of generating tumors upon (serial) intracranial transplantation that reflect the cellular heterogeneity of the parental tumor. For more lineage restricted cells with tumor initiating potential, the wider term tumor-initiating cell (TIC) should be used because it describes all cancer cells with tumorigenic potential [65].

### 2.1. Tumor Initiation and Cell of Origin

Neural stem cells (NSCs) are most active during neocortical development, where they give rise to distinct cell types and migrate to different regions in the developing brain, processes crucial for the formation of the nervous system [66]. The first compelling evidence for neurogenesis in the adult human brain came from a groundbreaking study using brain tissue obtained postmortem from patients who had been treated with bromodeoxyuridine (BrdU), demonstrating the adult human brain does generate new neurons from dividing progenitor cells, even at over 70 years of age [67]. In the mature brain, adult neurogenesis largely depends on NSCs, which generate lineage-restricted neural progenitor cells (NPCs) that are capable of self-renewal and differentiation into multiple lineages [68]. NSCs also give rise to oligodendrocyte precursor cells (OPCs), which comprise another pool of self-renewing cells that generate myelin-forming oligodendrocytes [69].

#### 2.1.1. NSCs, Committed Precurser Cells, and the Subventricular Zone

NSCs and their progeny can be found in the subgranular zone (SGZ) of the hippocampal dentate gyrus, striatum, frontal and temporal cortex, as well as the subcortical white matter, but the largest reservoir of these cells seems to reside in the subventricular zone (SVZ) [54,70,71,72,73,74]. In addition, NSCs of the SVZ share common characteristics with GSCs, as both exhibit elevated Nestin expression, increased motility, proliferative potential, pluripotency, and proximity to blood vessels [75]. In particular, there is extensive association and crosstalk between NSCs or GSCs and endothelial cells (and additional constituents within the highly vascularized SVZ, such as pericytes, astrocytes, and the extracellular matrix (ECM)), indicating that the SVZ provides a niche supporting maintenance of NSCs and GSCs in the brain (discussed in [76,77]). This is in line with the observation in GBM that GSCs are maintained within vascular niches [78,79]. Importantly, GSCs contribute to efficient vascularization within their niche by GSC differentiation into pericytes [80] or endothelial-like cells (GdECs) [81]. These findings indicate a crucial role for the SVZ in maintaining NSCs and likely provide a niche for GSCs. Therefore, it seems conceivable that GBM—and possibly additional brain tumors—originate from pluripotent progenitor cells within such niches in the SVZ [75,82].

Studies in mice suggest that dormant NSCs can be activated to proliferate in response to certain stimuli [83]. Monitoring NSCs in the SVZ in mice from infancy to advanced age together with single-cell RNA sequencing found that the ability to activate quiescent NSCs declines over age [84]. Still, activation of dormant NSCs is certainly one mechanism to initiate brain tumors in mice because these quiescent NSCs, once activated, can give rise to brain tumors in mouse models of medulloblastoma [85] and GBM [86].

Whether activation of quiescent NSCs also contributes to brain tumor formation in humans remains to be elucidated, but a key role for NSCs in this process appears to have been established. Genetic evidence for a crucial role of NSCs in the initiation of human GBM comes from a recent study using laser microdissection and single-cell sequencing to compare tumor free SVZ tissue with GBM tissue [87]. This study revealed a clonal relationship of driver mutations between cells in the SVZ and GBM tissue. Using genome-edited mouse models, the authors further demonstrated that NSCs, carrying driver mutations, migrate from the SVZ into distant brain regions, where they develop high-grade malignant gliomas [87]. Network modeling further supported this key role for NSCs and OPCs in GBM initiation because gliomas are found at strongly elevated frequencies at “hub regions” that are expected to be populated by NSCs and OPCs [88]. It has been known, for some time, that NPCs have the capacity to initiate tumors because Nestin-expressing NPCs are prone to AKT- and Ras-driven oncogenic transformation promoting gliomagenesis, while differentiated astrocytes are not [89]. In line with this, another study investigating the tumor-initiating potential of adult neural populations at various stages of lineage progression revealed that the tumorigenic potential decreases with increasing lineage restriction, corroborating that GBM initiation is facilitated by early mutational events in NSCs [90].

#### 2.1.2. Cell of Origin in IDH-Mutant and H3K27-Altered Gliomas

Further evidence for NSCs at the apex of a hierarchy of tumor initiating cells comes from single-cell RNA sequencing analyses. Single-cell profiling indicated that all isocitrate dehydrogenase (IDH)-mutant gliomas (astrocytomas and oligodendrogliomas), while differing in their tumor microenvironment (TME) and nonproliferating astrocytic and oligodendrocytic lineages, share a subpopulation of proliferative undifferentiated neural stem-like cells [91,92]. A similar picture of developmental cellular hierarchies also emerges for H3K27M gliomas, where OPC-like cells seem to be sitting at the apex [93]. Thus, these gliomas (IDH-mutant and H3K27M) each share glial lineages and developmental hierarchies, lending support to the cancer stem cell theory that gliomas may originate from a common and highly undifferentiated progenitor. The most likely candidates as possible cells of origin are NSCs and OPCs because both are known for their glioma initiating capabilities [75,82,89,94]. Remarkably, in the mature mouse brain, disruption of the same tumor suppressor genes in neural or oligodendroglial progenitor cells initiates two different GBM subtypes, indicating that the cell of origin is an important determinant of the GBM phenotype—at least in adult mice [95]. Taken together, these studies indicate a GBM cell of origin hierarchy, in which NSCs at the apex (possibly in the SVZ) exert key roles in GBM initiation. Finally, more lineage-restricted NPCs and OPCs may play decisive roles in the initiation of particular GBM subtypes.

## 3. Tumor Heterogeneity

Tumor heterogeneity is one of the most important hallmarks of GBM. Intertumor heterogeneity mostly refers to distinct alterations that can be found in individual tumors originating from the same organ. In GBM, intertumor heterogeneity is accompanied by intratumor heterogeneity on the genotypic and phenotypic level. Intratumor heterogeneity describes diversity within individual tumors. Hence, due to extensive intratumoral heterogeneity, a single GBM can exhibit numerous populations of cells, each with distinct alterations on the molecular level, which allow for an evolutionary advantage in overcoming selective pressures and microenvironmental cues imposed after radiation- and chemotherapy.

### 3.1. Intertumor Heterogeneity and GBM Classification and Subtypes

Initially, tumor-to-tumor variability in gliomas was assessed based on morphological appearance and histological differences between tumors. Still, identification of histological features (e.g., necrosis, vascular proliferation, cellularity, and mitotic figures) has not served as an efficient prognostic tool, nor has it allowed development of efficient treatment regimens to improve clinical outcome for GBM patients [96]. Hence, in 2016, the WHO updated guidelines and subdivided GBMs based on the mutational state of isocitrate dehydrogenase (IDH) genes due to the significantly worse outcome of IDH1/2 wild-type GBMs [97]. In 2021, the WHO further introduced significant changes by isolating IDH mutant gliomas into entities separate from GBM, highlighting IDH wild-type GBMs as the most aggressive adult-type gliomas [98]. Although histological analysis and mutational IDH status remain essential in the diagnosis of GBM, additional genomic studies have provided molecular classification as a newer tool, which has greatly advanced our understanding of these tumors. Initially, comprehensive patient sample analysis allowed for the subclassification into four different GBM subtypes [99], but single-cell transcriptome analysis confirmed only three subtypes in IDH wild-type GBM, which are designated as classical (CL), proneural (PN), and mesenchymal (MES) [34]. Amongst numerous genetic alterations, the most striking changes in gene expression used to define each subtype are found in platelet derived growth factor receptor alpha (PDGFRA)/IDH and p53 mutations for PN, epidermal growth factor receptor (*EGFR)* for CL, and neurofibromatosis type 1 (*NF1)* for MES GBM [34,99]. A more recent study identified four main cellular states (and their intermediate hybrids), designated as neural-progenitor-like cells (NPC-like) with *CDK4* amplifications, oligodendrocyte-progenitor-like cells (OPC-like) with *PDGFRA* amplifications, astrocyte-like cells (AC-like) with *EGFR* aberrations, and mesenchymal-like cells (MES-like) with Chr5q deletions and *NF1* alterations [41]. The preponderance of particular cellular states suggests the AC-like phenotype corresponds to the CL subtype, MES-like cells to MES, and a combination of OPC-like and NPC-like cells corresponds to the PN subtype. Recent single-cell transcriptomic approaches have provided additional insights into the complexity of GBM and identified overlapping, but also distinct, phenotypic states, such as an injury response state [100] and a variety of metabolic states [101]. Up to date, the classification methods proposed by Verhaak et al. [99] and Wang and coworkers [34] are widely used (reviewed in [102]), together with the plasticity model of hybrid cellular states from Neftel et al. [41]. There is association of certain subtypes with *IDH* status and CpG island methylator phenotype (G-CIMP) [76]. G-CIMP is tightly associated with *IDH* mutations (and also methylation of the O^6^-methylguanine-DNA methyltransferase (*MGMT*) promoter) and a more favorable prognosis. These gliomas are transcriptionally grouped in the PN subtype and represent “secondary” GBMs, as they evolve from lower-grade gliomas. Non-G-CIMP tumors are associated with the *IDH* wild-type, arise de novo as “primary” GBM, and can be grouped into all three subtypes (PN, MES, and CL) with worse prognosis as compared to G-CIMP/IDH mutant PN GBMs [76,103]. While transcriptional subtyping and epigenetic profiling of GBMs allows for better prediction of GBM evolution and treatment responses, these insights have not yet resulted in subtype selective targeted approaches that improve clinical outcome.

### 3.2. Intratumor Heterogeneity

Genetically, GBMs are mosaic tumors that contain multiple alterations at the genomic level, such as amplification of different receptor tyrosine kinase (RTK) genes within the same tumor, albeit with local variations resulting in spatial heterogeneity [30,32,36,104,105]. Additionally, recurrent tumors at distant brain sites exhibit a genomic profile that is highly divergent from the initial tumor, indicating evolutionary (i.e., temporal) heterogeneity [37]. Together, this genetic spatiotemporal heterogeneity results in highly diverse drug responses [31].

Additionally, on the transcriptional level, a patient’s tumor frequently exhibits a variety of cells representing multiple GBM subtypes. In fact, the term multiforme underscores the intratumoral heterogeneity of GBM. Molecular and single-cell analysis revealed that multiple GBM subtypes can coexist in the same tumor [35,106]. For instance, PN cells are found enriched in vascular areas, while hypoxic regions exhibit a more MES profile [107]. As a matter of fact, individual tumors can exhibit a spectrum of subtypes. To complicate this matter, cells within a tumor can adopt hybrid cellular states, meaning their transcriptional aberrations can be assigned to two or more subtypes at varying degrees [35,62,100,101]. Thus, while molecular subtyping allows for the designation of a dominant transcriptional program within a GBM, it does not represent the diversity of transcriptional subtypes within a tumor. This notion is of prognostic significance because tumors exhibiting increased heterogeneity are associated with decreased patient survival [35]. This is in agreement with a recent comprehensive study extending classical sequencing analysis with examination of proteome, phosphoproteome, acetylome, metabolome, and lipidome, which also reported shortened survival of patients with mixed subtypes [108]. In summary, intratumor heterogeneity might serve as a prognostic factor as (i) the predominant transcriptional program allows for some subtype dependent prognosis (i.e., median survival of 11.5 months in MES, 14.7 months in CL, and 17.0 months in PN [34]) and (ii) increased degree of intratumor heterogeneity (i.e., the proportion of tumor cells of alternate and hybrid states) is associated with decreased survival [35].

## 4. Plasticity

As discussed above, GBMs exhibit a high degree of heterogeneity [28,30,32,34,35,36,39]. Therefore, preexisting individual resistant clones in the initial patient tumor may occupy the space left by eradicated non-resistant cancer cells, which could explain the development of therapy resistance after treatment. However, genetic studies found that sister clones exhibit different resistance levels, and this is associated with variations in the gains of specific genes (e.g., *SOX2*) [109]. There is an abundance of continuous phenotypic gradients among resistant sister clones that cannot be explained solely by preexisting heterogeneity, but, rather, suggest acquired alterations account—at least in part—for the evolution of resistant clones [109]. Additional single-cell RNA analyses support this notion because GSCs can give rise to progeny residing on an axis from PN to MES gene signatures with varying degrees [110]. Finally, recent single-cell lineage tracing experiments combined with cellular barcoding revealed that GSCs of a certain state can give rise to tumors, exhibiting multiple cellular states, strongly supporting the concept of GSCs being responsible for the extraordinarily high degree of plasticity observed in GBM [41]. Together, these observations strongly suggest that GSCs are not static, but they exhibit fluid cellular phenotypes that can give rise to highly variable progeny, covering a spectrum of different transcriptional states over time and in space [40]. This could explain how, over the course of GBM progression, a tumor can switch from its predominant subtype to another one [33,34,111]. This shift seems to be further stimulated by radiation or chemotherapy, triggering mostly a proneural-to-mesenchymal transition of GSCs, which eventually results in chemo- and radioresistance [109,112]. Another layer of complexity was added by a recent study reporting that, besides activation of a developmental program, GSCs may alternatively activate a regenerative injury response program that may be mediated in part by cytokine signaling [100]. As pluripotent brain CSCs can give rise to multiple progenies with distinct transcriptional profiles, they have been the focus of investigation as potential drivers for this spatiotemporal evolution that may contribute to subtype switch and therapy resistance.

It seems established that GSCs are highly plastic, as GSCs of a certain subtype can give rise to hybrid progeny that eventually evolves into a different—and possibly more resistant-subtype. However, non-stem tumor cells may also exhibit some plasticity, allowing for dedifferentiation into more pluripotent progenitor cells [113]. Initial studies monitoring pluripotency marker expression demonstrated that marker negative cells (e.g., CD133^-^ cells) can initiate tumors that exhibit marker positive (CD133^+^) cell populations [49,50,114,115]. More importantly, when distinct subpopulations of patients’ GBM cells, purified for a given phenotypic state by multicolor flow cytometry, were transplanted into mice, they did not maintain their original phenotype. Instead, they were able to recreate phenotypic heterogeneity [116]. These alterations were stimulated by the environment because all sorted subpopulations increased CD133 expression under hypoxia, and this effect was reverted when switched to normoxic conditions [116]. This indicates that GBM cells represent different cellular states that respond to environmental cues, thus triggering plasticity that strives toward a phenotypical equilibrium of heterogenic cell populations within a given environment (Figure 1). This model includes the possibility that tumor cells also exhibit some plasticity and suggests that a GBM tumor depleted of its GSC population may be able to recreate heterogeneity by dedifferentiation of non-GSCs into more pluripotent progenitor cells [117]. This is in line with the observation that Temozolomide (TMZ) exposure stimulates conversion of differentiated tumor cells into GSCs that now express pluripotency markers and exhibit increased tumor initiating capabilities [118].

The reactivation of developmental programs (possibly further stimulated by radiochemotherapy, hypoxia, and other environmental triggers) promotes cancer cell plasticity in an attempt to maintain phenotypical equilibrium contributing to the highly heterogeneous nature of GBM. As this process is likely activated in cancer cells at varying stages of differentiation, a concept emerges in which GSCs can be generated de novo from bulk non-GSCs (which might be accelerated in the absence of pre-existing bona fide GSCs), and this needs to be considered when tailoring curative strategies that specifically target GSCs. An additional layer of complexity is added by the ability of GSCs to give rise to vascular pericytes [80] and endothelial cells [81], thereby possibly contributing to the formation or maintenance of highly vascularized stem cell niches that may provide a hub for GSCs.

In summary, plasticity allows not only transition of GSCs between distinct predominant subtypes (PN, MES, CL), but also confers the ability for GBM cells to transition between GSC and non-GSC states. GSC differentiation into endothelial cells and pericytes may remodel highly vascularized niches that provide a favorable environment for the evolution of additional pluripotent progenitor cells. Despite extensive recent advances on the molecular characterization of multiple GBM cellular states, each cell state can only be captured as a snapshot in time because anticipated plastic changes in cell behavior are—in large parts-inherently hypothetical. The notion that this high plasticity is driven by therapeutic challenges and environmental cues makes the development of curative approaches targeting the GSC population all the more challenging.

## 5. Therapy Resistance

GBM is a lethal brain tumor due to its ability to adapt to therapies. The most understood cause for its resistance to TMZ and ionizing radiation is the innate activation of DNA repair mechanisms [119]. However, recent studies, employing a high-complexity barcoding library, suggest that patterns of GBM recurrence (and resistance) exist on a spectrum between a priori fitness and a priori equipotency [120]. This is in line with previous reports, suggesting that GBM resistance can be stimulated by either preexisting chemo-resistant clones (linear model) [45] or alternatively by treatment induced changes in cell populations (divergent model) [121,122] (Figure 2). Additionally, tumor resection may act as a mechanistic stimulator that drives stem cell properties of residual cells enhancing self-renewal, therapeutic resistance and tumor recurrence [123]. Thus, GBM disease progression after chemoradiation may be driven by both innate and therapy-driven clonal dynamics and GSCs are key contributors to clonal expansion of therapy resistant subclones and GBM relapse.

### 5.1. DNA Repair Systems

TMZ and irradiation are essential components of the current multimodal standard-of-care therapy. Ionizing radiation generates irreversible clustered DNA damage, interstrand crosslinks, and both single- or double-strand breaks, while TMZ induces base pair mismatches. Thus, TMZ and ionizing radiation act by damaging DNA and are employed to trigger cell death. However, after therapy, GBMs inevitably recur, and this is largely due to resistant GSCs [124]. Early studies indicated a marked resistance of GSCs to chemotherapeutic agents, including TMZ [124,125]. The cytotoxic effects of TMZ are mediated mainly by methylating the O^6^-position of guanine in DNA, resulting in mismatches with thymine in double-stranded DNA (O^6^G-T), but the detoxifying enzyme O^6^-methylguanine-DNA-methyltransferase (MGMT) is able to remove these methyl residues. Increased *MGMT* promoter methylation restricts *MGMT* gene expression, resulting in enhanced sensitivity to alkylating agents, as TMZ is [126,127,128]. GSCs exhibit efficient DNA damage repair systems because CD133^+^ cells depict increased expression of MGMT, Breakpoint Cluster Region Pseudogene 1 (BCRP1), and anti-apoptotic proteins, which contribute to strongly increased TMZ resistance of CD133^+^ cells compared to their CD133^−^ counterparts [124]. While MGMT-negative GSCs were found to be sensitive to TMZ treatment [129], MGMT expressing GSCs were rather resistant because TMZ was unable to block their self-renewal capacity [130].

TMZ treatment results in an increased side population phenotype, indicative of an increased GSC population [131]. Hence, chemotherapy may stimulate the expansion of pre-existing drug resistant GSCs [45] or induce changes that promote resistance [121,122]. Therefore, TMZ treatment, while able to eliminate some GSCs, may selectively facilitate an increase in a subpopulation of resistant GSCs with retained self-renewal ability. Further, in *IDH*-mutated GBM, TMZ induces hypermutation of recurrent GBM [29,132,133], as evidenced by the specific TMZ-induced mutagenesis signature G>A/C>T [133,134,135]. Given that GSCs are resilient to TMZ mediated cell death [130], this raises the possibility that TMZ-induced mutations in GSCs may result in selective advantages that initiate recurrence of therapy resistant GBM. For instance, TMZ causes hypermutations, including gene alterations in components of the mismatch repair (MMR) system [42,78,134,136]. The MMR system can also recognize the O^6^-methylguanine produced by TMZ, but it additionally responds to additions, deletions, and base–pair mismatches. The MMR system detects unresolved lesions and induces double-strand breaks, resulting in replication fork collapse and subsequent cell cycle arrest and apoptosis [137]. Therefore, the MMR system is crucial for induction of cytotoxicity. Inactivating mutations in several components of the MMR system, such as the mutator L and mutator S homologs (MLH1, MSH2, MSH6, PMS2), are highly enriched in recurrent GBM samples from patients post TMZ treatment [33,138] and in patient-derived xenograft (PDX) lines that were allowed to develop resistance after repeated in vivo TMZ exposure [139].

In addition to MMR, single-stand break repair systems include base excision repair (BER) and nucleotide excision repair (NER). These systems recognize and remove the damaged portion of the DNA and then use the other strand as a template for repair [140]. While activation of BER and NER allows for DNA repair and thus survival, MMR is required to induce cell death after “futile repair cycle” [140]. A CRISPR-Cas9 screen using patient-derived GSCs corroborated a key role for a functional MMR in TMZ mediated cell death because TMZ-resistant clones were enriched for gRNAs targeting four core components of the MMR [141]. However, it appears that most GSCs are proficient in MMR responses [142], and alterations in MMR, BER, and NER have not yet been found to be enriched in GSCs, as compared to non-stem cells. Therefore, as far as these DNA repair systems are concerned, MGMT status seems to be the major determinant for TMZ resistance in GSCs.

### 5.2. DNA Damage Response (DDR)

In addition to MMR-induced double-strand breaks after TMZ treatment, irradiation is also mediating cytotoxic effects via potent induction of DNA-damage. A number of additional molecular mechanisms have been implied in GSC resistance to irradiation induced DNA-damage, such as the DNA-damage checkpoint [42] and, in particular, Ckh1/Chk2 [143], but also Notch [144], NF-κB [111], PARP [145], and EZH2 [146]. Together, these findings indicate that GSCs develop multiple mechanisms of resistance to DNA damage induced by TMZ and irradiation and therefore may require combinations of targeting agents. The role of DDR in GBM and TMZ- and radiation-mediated cytotoxicity has been extensively reviewed elsewhere [140,147,148].

### 5.3. Immunosuppression by Dynamic Crosstalk with Cells of the Immune System

Immunomodulatory approaches have revolutionized current cancer therapy and are already part of established treatment regimens in several solid tumors [149]. The successes in the treatment of other cancers have sparked great excitement in the neuro-oncology field, but expectations have been widely met with disappointment because all immunotherapies tested to date in unselected cohorts have failed to extend survival for GBM patients [150,151,152]. However, some patient subsets seem to benefit from immune checkpoint blockade (ICB) [153,154], possibly even in the recurrent setting [153,155], raising the possibility that improved identification strategies for the few patients with a likelihood to respond is urgently needed. One major obstacle limiting progress for ICB approaches is the dynamic crosstalk between GBM cells and immune cells and, in particular, the immunosuppressive GBM microenvironment (reviewed in [156,157]). Direct roles for crosstalk between GSCs and components of the immunosuppressive microenvironment have been well documented. For instance, GSCs repress immune sensors to sustain stem-like properties by avoiding negative regulation by the immune system [158]. GSCs can be found in close proximity to myeloid-derived suppressor cells (MDSCs), where they recruit and activate MDSCs to drive immune suppression [159]. Additionally, M2 macrophage polarization is known to suppress immune cell regulated antitumor responses [160], and several lines of evidence have demonstrated that GSCs induce and recruit M2 type tumor-associated macrophages (TAMs) to support tumor growth [161,162,163]. At the same time, GSCs protect themselves from T cell mediated killing by the secretion of PD-L1 containing extracellular vesicles (EVs), thereby suppressing antitumor immunity [164]. In addition, GSC-derived EVs may indirectly promote T cell immunosuppression via stimulating monocytes to adopt a more MDSC-like phenotype [165]. Vice versa, immune cells also modulate GSCs. For instance, TAMs have been shown to stimulate GSC maintenance to support malignant GBM growth [166] and invasion [167]. Further, the tumor microenvironment, and in particular immune-enriched areas, modulate genetic and transcriptional heterogeneity of GBM [168]. Remarkably, GSCs were found to populate a transcriptional gradient, ranging from a neurodevelopmental to an inflammatory program [100]. This is suggestive of inflammatory cytokines and immune cell communications playing a role in GSC plasticity and resistance to ICB. This is in line with previous observations, indicating that the tumor microenvironment can govern the phenotype of GSCs, leading to increased plasticity [41]. Together, there is compelling evidence for a crosstalk between GSCs and cells of the immune environment with a strong tendency to promote a GSC niche, to benefit from immune cell interactions in order to avoid recognition and destruction by cells of the immune system. Therefore, the potential roles of GSCs in response, adaptation, and resistance to immune-based approaches need to be further investigated.

### 5.4. Chemoresistance in GSCs

Besides the aforementioned resistance mechanisms, there are additional alterations that mediate resistance in GSCs. These include multidrug resistance (including ABC transporter mediated efflux), alterations triggered by the hypoxic microenvironment, increased anti-apoptosis signaling, autophagy induction, and metabolic dysregulation, which have been reviewed elsewhere [169,170].

## 6. Targeting Therapy Resistance in GSCs

Despite substantial advances in our understanding of GBM resistance mechanisms, therapeutic strategies aimed to suppress therapy resistance so far have failed. For instance, combination therapy approaches that include O6-benzylguanine (O6BG), an MGMT inhibitor, did not produce clinical benefit [171,172]. GBM therapy resistance is likely mediated by complex multifactorial pathway alterations that are the result of highly dynamic tumor evolution, driven by inter- and intra-tumor heterogeneity, which in turn are in large part attributed to GSCs. Heterogeneity and therapy resistance might even be conferred by GSCs in a sex-dependent context [173]. In this review, we will focus on more recent advances in targeting GSCs. Additional GSC directed approaches that include targeting of pathway components with established roles in cancer stem cells such as Notch1, Sonic hedgehog (shh), VEGFR, STAT3, and autophagy have been reviewed elsewhere [170].

### 6.1. Targeting Self-Renewal and Resistance in GSCs

Initially, brain cancer stem cells were isolated and functionally characterized based on their expression of the cell surface marker CD133 [24,25]. Further studies revealed that CD133^+^ cells exhibit greatly enhanced resistance to the standard-of-care regimen [42,124]. Additionally, ectopic overexpression of CD133 stimulates self-renewal capacity and proliferation [174]. Given the undisputed—albeit not exclusive—role of CD133^+^ cells in self-renewal and therapy resistance in brain cancers, it is conceivable that targeting GSCs via CD133 might be a promising strategy. Indeed, antibody-based CD133 targeting approaches showed promise in animal models [175,176]. Additionally, pharmacological CD133 targeting by pyrvinium, an FDA-approved anthelmintic compound that targets CD133^+^ cells, reduced self-renewal and tumor-initiating capacity [174]. These results have been corroborated by similar observations from other laboratories [177,178]. Hence, targeting of key contributors to GSC plasticity may limit heterogeneity. In line with this, combined targeting of BMI1 and EZH1, both associated with GSCs and distinctively expressed in different regions and GBM subtypes, may have promise in targeting a variety of GSCs within distinct regions of GBM [107]. Additionally, pathways that drive self-renewal, survival, and plasticity can serve as druggable targets specifically in GSCs [179,180,181,182,183]. Together, these results warrant further investigation of specific GSC targeting approaches and their potential for combination with additional curative strategies.

GSCs frequently adjust their metabolic regulation to adapt to environmental changes and metabolic stresses, such as hypoxia or low glucose levels, which contribute to maintenance of GSC self-renewal and GBM biology [184,185,186]. When compared to bulk GBM tumor cells, GSCs exhibit differences in their metabolism, such as increased de novo pyrimidine synthesis, which supports GSC self-renewal, tumorigenesis, and increased resistance to metabolic stress [43,187]. Hence, pharmacological targeting of pyrimidine synthesis may disrupt GSC frequencies and self-renewal [188]. However, strategies tailored to target metabolic alterations in GSCs may induce adaptive shifts in metabolic dependencies in GSCs, and they likely will require combination with additional anti-GSC strategies that specifically synergize with these anti-metabolic approaches. Whether targeting components of the circadian clock that are involved in metabolic processes [189] may provide a strategy that is more resilient to GSC adaptation will require further studies.

Extensive research into GSCs has identified several targetable scaffolds involved in transcriptional and epigenetic regulation. WDR5 was identified as a key epigenetic regulator in GSC organoids, and pharmacological WDR5 inhibition suppressed several stem cell marker genes and reduced self-renewal and tumor initiation [190], indicating that epigenetic regulatory scaffolds may provide feasible targets specifically in GSCs. Additional efforts have identified RNA regulatory processes as new targetable vulnerabilities in GSCs because they represent key contributors for GSC self-renewal and survival, both biological processes that contribute to therapy resistance. For instance, the RNA editing protein adenosine deaminase acting on RNA1 (ADAR1) was found to be the major RNA editing enzyme dysregulated in GSCs, and elevated ADAR1 contributes to self-renewal capacity and GSC survival via regulation of the GM2 ganglioside activator GM2A. Loss of either ADAR1 or GM2A in GSC-derived PDX models translated into blockade of tumor formation and longer survival [191]. Additionally, the novel finding of PDGF regulating mitophagy (an autophagic process important for mitochondrial quality control) via methyltransferase-like 3 (METTL3) and optineurin (OPTN), specifically in GSCs, suggested a targetable PDGFR-METTL3-OPTN axis for GBM. PDGF triggers mRNA decay of the tumor suppressor OPTN via METTL3, thereby suppressing mitophagy and targeting of METTL3 reduced tumor growth in GSC derived PDX models [192]. These effects are mediated by METTL3 induction of N6-methyladenosine (m^6^A) modifications.

Another m^6^A modifier, ALKBH5, is stabilized by the deubiquitinase USP36 in GSCs, stimulating tumor growth and TMZ resistance [193], and this USP36–ALKBH5 relationship may provide another targetable scaffold in GSCs. Additional target-specific m^6^A modifications seem to contribute to GBM by regulatory mRNA processes specifically in GSCs, but not NSCs. This can be explained by the observation that the m^6^A reader YTHDF2 specifically stabilized MYC transcripts in GSCs, but not in NSCs [194]. MYC stabilization, in turn, stabilized expression of IGFBP3, a protein that supports GSC viability. Remarkably, targeting this YTHDF2-MYC-IGFBP3 axis using IGF1/IGF1R inhibitors dramatically reduced GSC viability without affecting NSCs. Together, these data indicate m^6^A modifications and associated pathways as druggable scaffolds specifically in GSCs. Both m^6^A modifiers METTL3 and YTHDF2 are regulated by Yin Yang 1 (YY1), a regulator of chromatin structure that also plays a role in transcriptional elongation by interacting with cyclin-dependent kinase 9 (CDK9) [195]. Remarkably, targeting the YY1-CDK9 complex triggered interferon (IFN) responses in GSCs, and combined CDK9/PD-1 targeting reduced immunosuppressive Treg cells and MDSCs, while increasing cytotoxic CD8^+^ T cells, indicating that targeting transcriptional elongation processes might rewire the GBM microenvironment, thereby enhancing ICB therapeutic strategies, in particular in the GSC population [195]. This suggests that GSC targeting approaches may exhibit enhanced synergistic effects when combined with immune-based anti-GBM strategies.

### 6.2. GSCs in the Context of Immunomodulatory Approaches

Immunotherapeutic strategies have demonstrated positive effects in terms of quality of life and survival for patients with brain metastases [196,197], but not for patients with GBM [150]. This might be attributed to the fact that neoadjuvant anti-PD-1 therapy, while able to increase infiltration of T cells, is unable to diminish the dominant immunosuppressive role of macrophages and monocytes that is intrinsic to the unique GBM tumor microenvironment [155]. Administration of antibodies directed against epitopes highly expressed on GSCs may be a means to circumvent the negative effects of the immunosuppressive environment. For instance, anti-CD47 antibodies readily showed promising effects in immunocompetent GSC mouse models [198]. Additionally, combining ICB with oncolytic viruses may counteract immunosuppressive effects by changing the GBM microenvironment and stimulating innate and adaptive antitumor immune responses, as was demonstrated for melanoma [199]. Using GSCs as vehicles to deliver viruses may also improve the immune response. Hence, GSC targeting approaches using oncolytic viruses that replicate in Nestin expressing GSCs are underway (identifier no. NCT03152318).

Promising results for chimeric antigen receptor T (CAR T) cells in hematopoietic cancers triggered research efforts to boost CAR T cell efficacy and led to the development of novel strategies, such as multivalent CAR T cells (to avoid immune escape in heterogeneous tumors) or synNotch-CAR T cells that exhibit improved selectivity [200]. CAR T cells may also represent a promising GSC targeting strategy for GBM because a comparative evaluation of CD133 targeting immunotherapeutic strategies suggested a remarkable efficacy for CD133-specific CAR T cells without triggering adverse effects on hematopoietic stem cells [201]. However, CAR T cell-based anti-CD133 targeting may require postoperative intracranial delivery in order to mitigate immunosuppressive effects and to induce durable responses [202]. Efforts employing CRISPR screening aimed to enhance CAR T cell potency led to the identification of several hits in CAR T cells and knockout of identified targets augmented CAR T cell therapeutic responses in GSC mouse models [203]. However, while self-renewing GSCs are potentially amenable to immunologic targeting, all GSC directed immune-based approaches have to overcome the challenge of an immunologically “cold” tumor, maintained by the highly immunosuppressive environment, including, TAMs, MDSCs, and Tregs [204], which contribute to T cell exhaustion. [205]. Whether approaches such as altering the cytokine milieu or combinations with small molecules will improve CAR T cell efficacy and overcome the suppressive microenvironment will require further studies [204]. Alternatively, identifying key targets on GSCs that confer resistance to CAR killing might be helpful for increasing CAR T cell-mediated cytotoxicity, specifically in the context of GSCs [203].

Finally, a novel and exciting strategy is the generation of B cell-based vaccines (B_Vax_). These B_Vax_ are believed to exhibit efficacy by their ability to produce antitumor antibodies while also activating effector T cells. While B cells can switch from anti- to protumorigenic phenotypes, recent results in syngeneic mouse models are encouraging because they suggest that B_Vax_ act as potent antigen presenting cells (APCs), activating effector T cells and additionally showing high efficacy when combined with the standard-of-care [206]. In addition, B_Vax_ produced tumor-reactive antibodies with potential therapeutic properties. This indicates B_Vax_ may represent potent components of both cellular and humoral immunotherapy. The effect of B_Vax_ on GBM heterogeneity and the role of highly plastic GSCs in therapeutic responses is an area of investigation that will be followed with great interest.

## 7. Conclusions

Patients with newly diagnosed primary IDH^WT^ GBM have one of the poorest prognoses among different types of cancer patients. So far, no systemic therapeutic regimen has been shown to robustly improve survival for recurrent GBM since the introduction of the Stupp protocol in 2005 [4]. Since then, it has become apparent that long-term GBM management will require targeting of GBM cells with self-renewal capacity. In recent years, advances in sequencing technology have provided a more detailed understanding of the genomic and transcriptional landscape of GBM, highlighting the contributions of GSCs in the development and expansion of therapy resistant subclonal populations. Still, these insights have not yet translated into successful curative applications in the clinic. A major obstacle remains the high plasticity of GSCs and the extensive crosstalk with the microenvironment, including immune cells. Mouse models of recurrent GBM indicate there might be a treatment “window of opportunity” at the minimal residual disease (MRD) state, when heterogeneity and plasticity are relatively low [120]. A proposed model for targeting highly plastic GBM is illustrated in Figure 3. Thus, MDS might represent a transient state in which GBMs are particularly vulnerable to approaches that combine GSC targeting, disruption of communication with the GBM environment, and immune-based approaches.

## Figures and Tables

**Figure 1 cancers-15-03458-f001:**
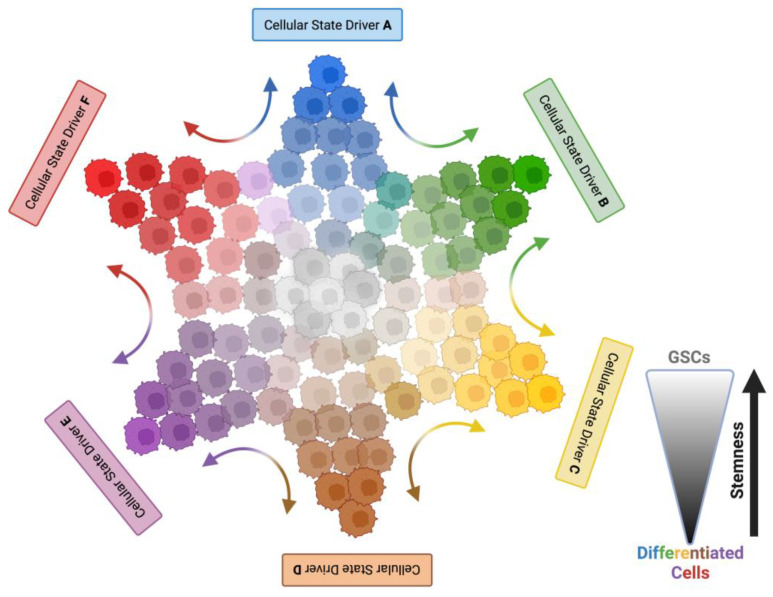
Simplified, two-dimensional model of dynamic adaptations of highly plastic GSC populations. GSCs (shaded in grey) reside at the center of GBM plasticity giving rise to multiple cellular states (indicated by differently colored GBM cells), thereby driving tumor heterogeneity. Cellular states (each represented by a different color) can be driven by intrinsic and extrinsic factors (e.g., therapeutic intervention, hypoxia, injury response, niche composition, genetic alterations etc., as indicated by “Cellular State Driver A–F”). Cells of a certain state can transition to another cellular state (indicated by dual-colored arrows). Plasticity also produces several intermediate hybrid states that share properties of two cellular states (indicated by cells with intermediate colors between two cellular states). Created with BioRender.com (accessed on 22 June 2023).

**Figure 2 cancers-15-03458-f002:**
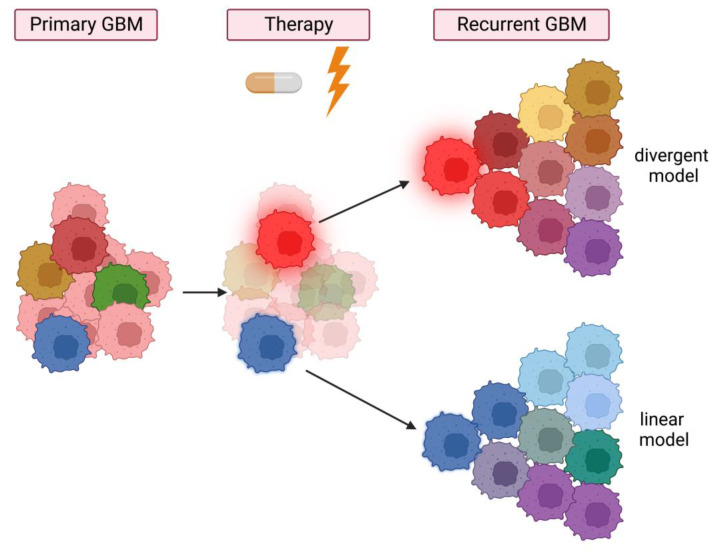
Model of GSC subclones driving therapy resistance and recurrence of heterogeneous GBM. Primary heterogeneous GBM (**left panel**) with GBM cells (rose cells) and plastic GSCs (colored cells). Administration of therapy (**middle panel**) provides an environment that promotes the selection of subclonal GSC populations, which either already are therapy resistant (blue cell; linear model) or acquire therapy-driven resistance (red cell; divergent model). The resulting GSCs are highly plastic and can seed tumor relapse of heterogeneous recurrent GBM (**right panel**). Created with BioRender.com (accessed on 17 March 2023).

**Figure 3 cancers-15-03458-f003:**
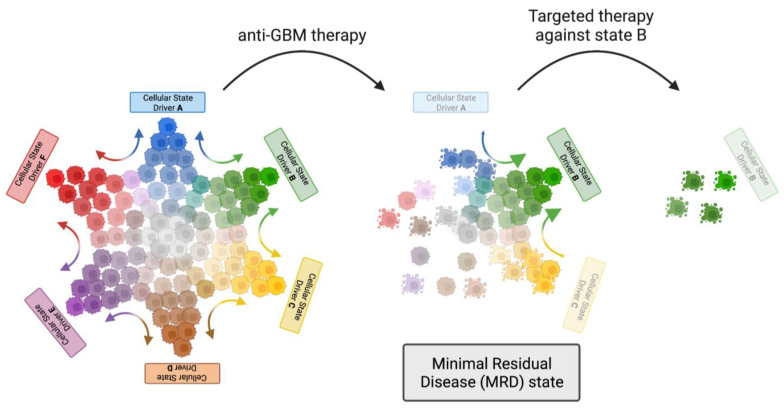
Proposed therapeutic approach to target highly plastic GSCs in GBM. Highly plastic GBM requires multimodal and sequential therapeutic approaches. The first treatment (anti-GBM therapy) eliminates some cellular states while also driving cells (including GSCs) towards one single state. This post-treatment minimal residual disease (MRD) state exhibits greatly reduced heterogeneity and is now vulnerable to strategies specifically targeting the remaining cellular state. Created with BioRender.com (accessed on 22 June 2023).

## Data Availability

No new data were created or analyzed in this study. Data sharing is not applicable to this article.

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
