# Peer review of "Emerging Role of Glioma Stem Cells in Mechanisms of Therapy Resistance"

_cancers, 2023, doi:10.3390/cancers15133458_

Round 1

Reviewer 1 Report

Eckerdt et al. made a comprehensive review with respect to the role of glioma stem cells (GSCs) in therapy resistance of gliomas, from the identification and characterization of GSCs to elucidating their contributions to GBM malignancy and mechanisms of therapy resistance. They also summarized novel strategies that specifically target GSCs. I have a few points that should be addressed before acceptance.

1.     The median overall survival of GBM has been extended to ~20 month with the addition of TTFields, the authors should update this.

2.     There is only one subtitle under the ‘2. Cancer Stem Cells in Glioblastoma’ section. I suggest adding 1-2 subtitles into this part since ‘tumor initiation and cell of origin’ is a big topic.

3.     In the 3.1. Intertumor Heterogeneity and GBM Classification and Subtypes, the authors did not mention the newest classification of gliomas (WHO-2021).

4.     The content of 5.2. DNA Damage Response (DDR) is relative short, actually DDR is a very important mechanism for TMZ- and radiation-mediated cytotoxicity and resistance.

5.     It seems that a figure/diagram is needed to visualize the main content of the review.

Author Response

We are pleased about this reviewer’s positive response, and we very much appreciate the reviewer’s comments and suggestions which we find extremely helpful.

  1. We updated this and included a reference indicating overall survival may extend to 20 month (page 2, lines 51ff).
  2. We added 2 more subtitles in the section “tumor initiation and cell of origin”.
  3. We have now included the newest GBM classification of the WHO-2021 (page 5, lines 215ff).
  4. We agree that DDR is an important topic. As this topic has been extensively covered in other review articles, we have listed three review articles (references 140, 147, 148; see page 9, lines 423f) that extensively cover this topic.
  5. We agree that a figure/diagram would be very helpful. Instead of producing a new figure, we now refer to the schematics of Jeremy Rich’s laboratory as illustrated in Prager et al., Trends Cancer 2020 (ref. 118). Together, the “attractor state model of glioblastoma” (Fig. 1) and the “therapeutic approach to glioblastoma stem cell adaptation and heterogeneity” (Fig. 3) provide elegant and comprehensive illustrations of plasticity induced adaptation of GSCs as well as potential anti-GSC therapeutic strategies.

Reviewer 2 Report

I congratulate the authors on a well written compendium on the role of GSC in glioblastoma resistance to therapy.

Author Response

Reviewer #2:

We thank the reviewer for this very positive assessment.

Reviewer 3 Report

The authors present a well-articulated review describing the state of affairs related to glioblastoma (GBM) therapy resistance with a focus on stem-like cells (i.e. GSCs) as key contributors of these phenotype. The authors presents a thoughtful and comprehensive commentary highlighting key aspects thought to contribute to the emergence of therapy-resistant cell populations in GBM on possible approaches to target them. Overall, a pleasant read that should be of interest to the GBM enthusiasts.

This reviewer has a couple of suggestions aimed at enhancing the manuscript:

1.The authors should consider including an additional illustration summarizing the different mechanisms highlighted in the text contribute to the emergence of resistant cell populations. 

2. As the authors correctly point out, despite our growing molecular understanding of GBM, it's been challenging to translate these new-found knowledge to the clinic. This reviewer would be interested in reading about the author's perspective on this and their thoughts about how to overcome these challenges. 

Author Response

We are pleased found our manuscript to be a pleasant read that represents a thoughtful and comprehensive commentary.

  1. We agree that a figure/diagram would be very helpful. Instead of producing a new figure, we now refer to the schematics of Jeremy Rich’s laboratory as illustrated in Prager et al., Trends Cancer 2020 (ref. 118). Together, the “attractor state model of glioblastoma” (Fig. 1) and the “therapeutic approach to glioblastoma stem cell adaptation and heterogeneity” (Fig. 3) provide elegant and comprehensive illustrations of plasticity induced adaptation of GSCs as well as potential anti-GSC therapeutic strategies.
  2. How to translate our growing molecular understanding of GBM to the clinic is a very interesting but also challenging topic. Given the high plasticity of GSCs and the ability of non-stem cells to dedifferentiate into stem-like cells makes this a very difficult topic. As every molecular assessment of GBM tumors can only represent a snapshot in time for this patient’s tumor, outcome prediction after GSC targeted therapy is highly speculative. In an effort to generalize this topic, we propose targeting a patient’s tumor at the minimal residual disease state (see page 13, 7. Conclusions). This strategy is also schematically illustrated in Fig. 3 of Prager et al., Trends Cancer 2020 (ref. 118).

Reviewer 4 Report

The review is a well-made comprehensive script made for Cancer Stem Cells (CSC) in specific for glioblastoma and makes an astounding effort to make it accessible for anyone in the field. I think is well thought out in the writing as it goes explaining the intricacies of cancer stem cells, and makes it clear to the now parading and its intricacy. It cites well and modern reviews in the field and actualizes the new landscape and opportunities in the field to progress

Author Response

We are pleased about this reviewer’s positive response, and the notion of our manuscript representing a well-made comprehensive script for GSCs that is accessible for anyone in the field.

Reviewer 5 Report

Frank Eckerdt et al. have provided a comprehensive summary of the current understanding of glioma stem cells (GSCs) and their crucial role in glioblastoma malignancy, as well as novel strategies for targeting GSCs in the context of immune-based therapies.

While this review is well-written and informative, there are a couple of points that the authors could consider to improve this review further:

1.    One important aspect that should be mentioned is the role of cell proliferation in therapy resistance. While it was previously thought that slow cell proliferation was responsible for GSCs' radio- and chemoresistance, recent single-cell analysis studies have shown that GSCs may have faster proliferation rates than non-GSCs. There is an ongoing debate in this area. Some researchers found that GSC proliferation dynamically changes during fractionated irradiation or chemotherapy, leading to resistance acquisition. They are important to further understanding the mechanism. Therefore, it would be beneficial to discuss this topic in the review.

2.    Although the review is well-summarized, only one simple figure is provided. It may not be informative enough for readers. It would be more impactful if one more figure illustrating the resistance mechanisms of GSCs and corresponding treatment strategies were included.

Author Response

We are pleased this reviewer finds we have provided a comprehensive summary of the current understanding of GSCs and their crucial role in GBM malignancy.

  1. Proliferation rate of GSCs has been an interesting topic that is still subject of debate. We have now included a small paragraph on this topic (see page 2, lines 85ff).
  2. We agree that a figure/diagram would be very helpful. Instead of producing a new figure, we now refer to the schematics of Jeremy Rich’s laboratory as illustrated in Prager et al., Trends Cancer 2020 (ref. 118). Together, the “attractor state model of glioblastoma” (Fig. 1) and the “therapeutic approach to glioblastoma stem cell adaptation and heterogeneity” (Fig. 3) provide elegant and comprehensive illustrations of plasticity induced adaptation of GSCs as well as potential anti-GSC therapeutic strategies.